# The Role of Ultrasound in Epicutaneo-Caval Catheter Insertion in Neonates: Systematic Review, Meta-Analysis and Future Perspectives

**DOI:** 10.3390/diagnostics13172850

**Published:** 2023-09-03

**Authors:** Vito D’Andrea, Valentina Cascini, Rosellina Russo, Alessandro Perri, Giorgia Prontera, Gina Ancora, Giovanni Vento, Gabriele Lisi, Giovanni Barone

**Affiliations:** 1Neonatology Unit, Department of Woman and Child Health and Public Health, Fondazione Policlinico Universitario Agostino Gemelli IRCCS, 00168 Rome, Italy; alessandro.perri@policlinicogemelli.it (A.P.); giorgia.prontera@libero.it (G.P.); giovanni.vento@unicatt.it (G.V.); 2Pediatric Surgery Unit, Hospital Spirito Santo, 65124 Pescara, Italy; valentina.cascini@gmail.com (V.C.); gabriele.lisi@asl.pe.it (G.L.); 3Department of Diagnostic Imaging, Oncological Radiotherapy, and Hematology, Neuroradiology, Fondazione Policlinico Universitario Agostino Gemelli IRCCS, 00168 Rome, Italy; rosellina.russo@policlinicogemelli.it; 4Neonatal Intensive Care Unit, Azienda Sanitaria Romagna, Infermi Hospital, 47923 Rimini, Italy; gina.ancora@auslromagna.it (G.A.); gbarone85@yahoo.it (G.B.); 5Pediatric Surgery Unit, Department of Medicine and Aging Science, “G. D’Annunzio University”, 66100 Chieti, Italy

**Keywords:** ultrasound, epicutaneo-caval catheter, neonates, tip location

## Abstract

Chest and abdominal X-rays after the insertion of an epicutaneo-caval catheter in infants are the standard method of checking the tip location in many neonatal intensive care units. The role of ultrasound in the tip location of the epicutaneo-caval catheter in neonates has been the subject of many recent studies. This systematic review investigates the accuracy of epicutaneo-caval catheter tip location by comparing ultrasound and conventional radiology. We performed a systematic literature search in multiple databases. The selection of studies yielded nineteen articles. The systematic review and meta-analysis were performed according to PRISMA (Preferred Reporting Items for Systematic reviews and Meta-analysis). The analyses showed that ultrasound is a better imaging technique for epicutaneo-caval catheter tip location in the neonatal intensive care unit than conventional radiology. By improving operator training and selecting a standardized echography protocol, ultrasound could become the gold standard for visualizing the epicutaneo-caval catheter tip in the neonatal intensive care unit. This would have some important benefits: (1) increased accuracy in tip location (2); a more rapid use of the central venous access (3); and a significant reduction in radiation exposure.

## 1. Introduction

The use of epicutaneo-caval catheters (ECCs) has become increasingly common in the Neonatal Intensive Care Unit (NICU) [1,2]; approximately 8.3–33% of neonates admitted to the NICU require an ECC insertion [2]. ECCs are mainly placed in neonates for the administration of drugs not suitable for the peripheral route and for middle-term parenteral nutrition, especially in preterm infants, in neonates with feeding difficulty or feeding contraindicated, such as being small for gestational age (SGA) with abnormal antenatal Doppler [1,3]. ECCs are often called peripherally inserted central catheters (PICC), which makes sense because they are central venous access devices and because they are inserted in peripheral, superficial veins. However, the World Congress on Vascular Access (WoCoVA) Foundation—which is the recognized global network of the associations of vascular access—has recently recommended a new terminology of central venous access; according to WoCoVA, a central venous catheter (CVC) is defined as a central venous access device placed by cannulation of the deep veins of the arm (PICC), deep veins of the supra/infraclavicular area (centrally inserted central catheter (CICC)) or deep veins of the groin (femorally inserted central catheter (FICC)). Deep veins are defined as veins that lie deeper than 7 mm from the surface of the skin. This implies that ECCs are not PICC. They represent a different device, offer different performances and have a different complications rate [4].

Complications related to ECC insertion include catheter-associated sepsis (CLABSI), thrombosis, catheter occlusion and malfunction, arrhythmias, phlebitis, pleural and pericardial effusion [1].

The optimal tip position of an ECC is essential in minimizing most of the complications. An ECC in a central position had a significantly lower complication rate than those with the tip in a peripheral vein [1,3]. Due to their small caliber, ECCs are also prone to secondary malposition (so-called tip migration). In fact, breathing, arm movements and changes in body position may be associated with movements of the tip upwards or downwards [5,6]. Moreover, as the premature neonate grows in body length, the tip of the catheter will move progressively upward [7]. Secondary malposition due to late tip migration is reported in 30–35% of cases [8]. According to Srinivasan et al. [9], in the first 24 h after insertion, 47% of ECCs placed in the upper extremities migrate to a non-central position, with 32.6% migrating toward the heart. When the position of the arms changes, the tip may move an average distance of 2.2 rib spaces and a maximum distance of 3.5 rib spaces [10].

Therefore, a proper method for the assessment of tip location during and after ECC insertion is highly recommended [4,6,8]. Current strategies for avoiding primary and secondary malposition do not seem to be effective [11], although there is consensus that an optimal tip position is essential for the clinical efficacy and safety of ECCs [3]. In this regard, ultrasound use is very interesting since it offers a real time visualization of the tip allowing the possibility to constantly locate the tip of the catheter in virtually every clinical situation.

Several methods have been used to establish the tip location of ECCs. Currently, chest/abdominal radiography (X-ray) remains the most common technique. Unfortunately, however, it is not accurate, is a post-procedural methodology, and exposes neonates to ionizing radiation [11]. Furthermore, these catheters are not always placed at the optimal position during the first attempt; therefore, the use of a post-procedural technology for tip location quite often implies the repositioning of the ECC, which can cause several complications such as CLABSI, as well as contributing to delays in care, increasing the exposure of ionizing radiation and increasing overall procedural time [3]. A national survey about central venous access devices (CVAD) in the NICU showed that this radiological method is still used in 82% of cases [4,12]. The radiological assessment is based on the relationship between the projection of the tip and several anatomical landmarks, such as the diaphragm, the cardiac silhouette and the vertebral bodies [4]. A single anteroposterior (AP) chest radiograph is the most used technique; in some cases, a lateral chest X-ray is also added. Recently, several studies have questioned the accuracy of X-rays in verifying the correct tip position both in UVCs and in ECCs [13]. The cavo-atrial junction (CAJ) is esteemed to be located approximately two vertebral bodies below the carina [14]. After estimating the location of the CAJ, for an ECC inserted in the upper extremities, the tip position should be 1–2 cm above the right atrium in term and 0.5 cm in preterm neonates [13], or 2 cm outside the cardiac silhouette in term and 1 cm in preterm neonates [8]. This method is not accurate because the CAJ cannot be seen on the radiography, but only guessed by radiological landmarks.

The INS guidelines 2021 recommend the use of intraprocedural methods to identify a catheter’s tip during the insertion procedure [5]. In recent years, IC-ECG has been used to assess the correct position of catheter tips in adults and children, during PICC, CICC or FICC insertion [7]. This method avoids X-ray exposure, and it is simple, safe and highly accurate. Primary malposition does not occur, and the costs and complications of repositioning are avoided [15]. However, due to the small caliber of the catheters, IC-ECG may be difficult or not feasible when applied to ECCs. However, several studies on ECCs showed that IC-ECG is nevertheless more accurate than X-ray [3,7,15,16].

Ultrasound/echocardiography (US) has been recommended by the last version of the INS guideline, especially in neonates, as the standard technique for tip location. US can be used in real-time, are reliable and offer a dynamic assessment of the tip position with other advantages like no radiation exposure, minimal handling of the neonate, the identification of secondary malposition due to tip migration and early diagnosis of catheter-related complications [13].

The aim of the present paper is to perform a systematic review and meta-analysis about the methodology of tip location for ECCs and give the readers a focus on future research.

## 2. Materials and Methods

This systematic review and meta-analysis were performed according to PRISMA (Preferred Reporting Items for Systematic Reviews and Meta-analysis) [17] (Figure 1).

Using a stated search strategy (see inclusion and exclusion criteria), two investigators (VC, VDA) individually screened the main databases (PubMed, Embase, Cochrane library and Web of Science) with combined keywords to evaluate the articles about tip location in ECCs using X-ray and US in neonates, updated to May 2022. Only papers in the English language were included. MeSH headings and terms used were “ultrasound”, ”ultrasonography”, “point-of-care ultrasound”, “echocardiography”, “infant”, “neonate”, “newborn”, “central catheter”, “PICC”, “peripherally inserted central catheter”, “ECC”, “epicutaneo-caval catheter”. In addition, the reference lists of relevant reviews were manually searched to obtain additional articles.

### 2.1. Inclusion Criteria and Exclusion Criteria

Eligible studies included in this meta-analysis followed these criteria:Prospective, retrospective observational studies and clinical trials regarding the study of ECC tip position in neonates and infants;Studies comparing two procedures about tip position analysis, such as US or echocardiography and standard X-ray;

Articles excluded were case reports and reviews or studies without valid data about the comparison of these two techniques.

The full text of theoretically suitable papers was retrieved and individually assessed for eligibility by the same two authors (VC, VD) that performed data extraction. Any divergence over the entitlement of papers was solved through a further debate with a third author (GLI).

### 2.2. Data Analysis

The statistical analysis was performed using RevMan 5.4 (The Cochrane Collaboration, Copenhagen, Denmark).

The risk ratio (RR) was assessed for categorical variables. Differently, mean differences (MD) were preferred in the case of continuous variables. Both results were reported with 95% confidence intervals (CI). Data were expressed as mean ± SD.

The random or fixed-effects model was applied depending on heterogeneity. Heterogeneity among the included studies was assessed by I^2^ test. When I^2^ < 50%, the fixed model was used. Quantitative and demographic data were compared using Fisher’s exact test and expressed as number, percentage or mean ± SD using relative risk (RR) and 95% CI. A *p* < 0.05 was considered significant.

### 2.3. Quality Assessment

Two authors (VD and VC) assessed the risk of bias for individual studies. This assessment was achieved with a methodological index for nonrandomized studies (MINORS) [18]. Dissimilarities between the two authors (VD and VC) were solved through a discussion with a third author (GB). The score for this index ranges between 0 and 24 points. The “gold standard” cutoff was 19.8 points (Table 1). With regard to the quality of each outcome, we graded the quality of evidence, thanks to the Grading of Recommendations Assessment, Development and Evaluation (GRADE) methodology [19].

The quality of evidence was graded as high, moderate, low and very low in all results. Observational studies were assessed as low quality of evidence. The quality of evidence was further reduced in the case of risk of bias, inconsistency, indirectness imprecision and publication bias. MINORS was adopted to judge the risk of bias in observational papers. Inconsistency was determined according to heterogeneity, and I^2^ value was used to evaluate heterogeneity. As established in the Cochrane guidelines, heterogeneity was assessed as low, moderate, substantial and considerable when I^2^ values were 0–40, 30–60, 50–90 and 75%–100%, respectively) [20]. If a score overlapped two groups, we inserted a mixed inconsistency (e.g., low/moderate) in our GRADE table (Table 2).

**Table 1 diagnostics-13-02850-t001:** Risk of bias assessment for individual studies using methodological index for nonrandomized studies (MINORS).

Item	Jain [1]	Motz [11]	Katheria [21]	Saul [22]	Zaghloul [23]	Johnson [24]	Oleti [25]	Ohki [26]	Tauzin [27]	Brissaud [28]	Ren [29]	Kuschel [30]	Telang [31]	Madar [32]	Diemer [33]	Shabeer [34]	Huang [35]	Motz [36]	Grasso [37]
1. A clearly stated aim	2	2	2	2	2	2	2	2	2	2	2	2	2	2	2	2	2	2	2
2. Inclusion of consecutive patients	2	2	2	2	2	1	2	2	2	2	2	2	2	2	2	2	2	2	2
3. Prospective collection of data	2	2	2	2	2	0	2	0	2	2	0	0	2	0	0	2	2	2	2
4. Endpoints appropriate to the aim of the study	1	2	2	2	1	2	2	2	2	2	2	2	2	2	2	2	2	1	2
5. Unbiased assessment of the study endpoint	1	1	1	1	1	1	1	1	1	1	1	1	1	1	1	1	1	1	1
6. Follow-up period appropriate to the aim of the study	0	1	0	0	0	0	0	0	0	0	0	0	0	0	0	0	0	0	0
7. Loss to follow-up less than 5%	0	0	0	0	0	0	0	0	0	0	0	0	0	0	0	0	0	0	0
8. Prospective calculation of the study size	2	2	2	2	2	0	2	0	2	2	0	0	1	0	0	2	2	2	2
9. An adequate control group	2	2	2	1	2	0	2	1	2	2	2	2	2	0	1	2	2	1	2
10. Contemporary groups	2	2	2	2	2	1	2	1	2	2	2	2	2	1	2	2	2	2	2
11. Baseline equivalence of groups	2	2	2	2	2	2	2	2	2	2	2	2	2	2	2	2	2	2	2
12. Adequate statistical analyses	0	0	2	0	2	0	2	0	2	2	0	1	1	0	0	2	2	0	1
Total score	16	18	19	16	18	9	19	11	19	19	13	14	17	10	12	19	19	15	18

0 = not reported; 1 = reported but inadequate; 2 = reported and adequate. Validated “gold standard” cut-off: 19.8.

GRADE Working Group grades of evidence:

High quality: Further research is very unlikely to change our confidence in the estimate of effect.

Moderate quality: Further research is likely to have an important impact on our confidence in the estimate of effect and may change the estimate.

Low quality: Further research is very likely to have an important impact on our confidence in the estimate of effect and is likely.

Very low quality: We are very uncertain about the estimate.

## 3. Results

Initially, 296 articles were identified (PubMed, Embase, Cochrane library and Web of Science). The relevant articles and publications (91) were selected in two stages. During the first stage, the titles and abstracts of the articles were screened, and non-relevant articles were excluded (71). Of papers that were deemed relevant to the study objectives, 19 met the inclusion criteria and were included in the analysis [1,11,21,22,23,24,25,26,27,28,29,30,31,32,33,34,35,36,37]. Among these articles, 15 are prospective observational studies [1,11,22,23,26,27,28,30,31,32,33,34,35,36,37], two retrospective papers [24,29] and two randomized controlled trials [21,25].

A total number of 1094 neonates were included, with 1038 ECCs inserted. Eleven papers, with 1030 ECCs, reported the site of insertion: in 55% (566/1030) the ECCs were located in the upper extremities; in 45% (464/1030) in the lower extremities [11,21,25,26,29,30,31,34,35]. Ten of these studies, including 790 neonates, reported the gender of patients: 61.5% (486/790) were male and 38.5% (304/790) were female [1,11,21,27,29,30,31,35,36,37].

Mean gestational age (GA) was 28.1 ± 4.0 [1,11,21,22,24,25,28,29,30,31,32,34,35,36] and 61.6% (318/516) were preterm infants [1,11,29,34,35,36] Mean birth weight was 1299 ± 320.8 [1,11,21,22,23,24,25,28,29,30,31,32,34,35,36,37], with 69% (361/520) of neonates weighing < 2500 g [11,23,24,25,27,29,36,37].

### 3.1. Outcomes

#### 3.1.1. Tip Visualization

The total tips visualized at US were 78%, whereas tips visualized at X-ray were 76.6%, *p* = 0.01, RR 0.88 [0.79–0.97] (Figure 2). These data were reported in 17 papers [1,21,22,23,24,25,26,27,28,29,30,31,32,33,34,35,36]. The overall percentage of agreement of tip location between X-ray and US was 89% [1,11,24,27,31,36,37].

#### 3.1.2. Correct Tip Position

The comparison between X-ray and US in analyzing correct tip position at CAJ was reported in 17 papers [1,11,21,22,23,24,25,26,28,29,30,31,32,33,34,35,37]. The tip of the catheter was properly visualized at US in 87%, while at X-ray study in 77%, *p* = 0.002, RR 0.74 [0.60–0.90] (Figure 3).

#### 3.1.3. Total Malposition

Overall malposition was 30% (267/986) and was analyzed in 17 papers [1,11,21,22,23,24,25,26,27,28,29,30,31,32,33,35,37]. The role of US compared to X-ray in the visualization of malposition was analyzed in detail in 15 papers [1,11,21,22,24,25,26,27,28,29,31,32,34,35,37]. US revealed 85.5% instances of malposition, whereas X-ray showed 22%, *p* = 0.04, RR 0.13 [0.02–0.23] (Figure 4).

#### 3.1.4. Timing of Insertion and Securing ECC

Only four studies reported these data [21,28,34,35]. The mean time to insert and secure ECCs during US tip navigation/location was 33.62 ± 15.74 min vs. 88.80 ± 21.05 min required for X-ray, *p* < 0.07 (Figure 5). The main acoustic views used for tip navigation/location were subxiphoid short and long-axis views, apical four chambers view, parasternal long- and short-axis views. In seven papers, these data were not reported [22,24,26,28,30,31,32].

#### 3.1.5. Saline Bolus

In six papers, the US tip location is amplified by a flush of 0.5–2 mL of saline solution [21,24,29,30,31,37], for a total of 285 ECCs inserted. Tips visualized were 91.6% vs. 74.3% of visualization without the use of saline bolus, *p* < 0.0001, OR: 1.232 [1.163–1.305] (Figure 6).

## 4. Discussion

One of the main issues in ECC insertion is the correct tip position. In fact, catheter malposition increases the risk of complications, such as CLABSI, described in 20% of neonates, and thrombosis/thrombophlebitis [5,7], reported in up to 10% of VLBWs with a central line [38]. When the tip of an ECC is not in a central position (outside the SVC, CAJ or IVC), this will increase the risk of thrombosis by 4.5 times [39]. On the other hand, when the tip is located inside the heart there is an increased risk of erosion, cardiac tamponade and arrhythmias [5,40]. For this reason, the 2021 INS guidelines reported that the correct tip location is the CAJ, the same as for ECCs [5,38].

In the literature, the incidence of ECC-related complications varied from 2.9 to 49.5% [1,26,41], leading to a high rate of non-elective removal of the catheter in up to 43.27% of cases [11,42].

Thus, a proper technique of tip location is crucial during the placement of an ECC [4,6,8].

However, the common strategy for tip location used in most NICUs (also known as conventional radiology) is not reliable [11].

In fact, X-ray is not accurate since it infers the position of the tip using anatomical landmarks, which are essentially based on statistical data [28]. Furthermore, plain radiography exposes the neonates to ionizing radiation, which might have long-term effects [28].

Although the use of sonography is advocated in the NICU to localize device position, it has not yet been routinely employed [22]. This is probably for two reasons: (1) the lack of a standardized protocol for tip location (2); the lack of standardized US training.

During the fulfillment of the present work, the studies were carefully evaluated to include all possible studies on this topic. In fact, even though the present meta-analysis refers to ECCs in newborns, during the term selection, the word PICC was also included since the utilization of the term as referring to ECCs in neonates is very common and is unfortunately a constant source of confusion in the scientific literature. The ECCs that are inserted in neonates are completely different to PICCs, which are used in children and adults. ECCs are small bore catheters (1–2.7 Fr) composed of silicone or old-generation polyurethane, inserted via superficial veins of the limbs or scalp using direct vein visualization. PICCs are larger catheters (3 Fr and more) composed of new-generation polyurethane, usually power-injectable, and inserted into the deep veins of the arm (brachial, basilic, axillary) using ultrasound guidance. There is a huge technological leap between these two devices, which translates in different performances: PICCs are appropriate for blood sampling, high flow infusion (up to 1 mL/s vs. 1 mL/min of ECC), hemodynamic monitoring (central venous pressure, central venous sampling for oxygen saturation in mixed venous blood, etc.) and infusion of blood products; they have extended dwell time (even months); they can be secured with subcutaneously anchored sutureless systems, thus abolishing the risk of dislocation; their tip can be safely located using intracavitary electrocardiogram (ECG; difficult to use for ECCs); and an accurate diagnosis of PICC colonization or infection is consistently possible by the delayed time to positivity (DTP) method, which is not applicable to ECCs. All peripheral central catheters inserted in newborns are ECCs, and this word should be the only term used. However, in order to include all relevant studies, we also looked for the word PICC during our research; nevertheless, only the studies that actually referred to neonates (and therefore to ECCs) were indeed included in the analysis.

Our data showed that the ability to successfully locate the tip is 78% for US and 76.6% for X-ray, with a significant difference between the two techniques, as shown in Figure 2. However, these raw data about the performance of US are underpowered for the reasons mentioned earlier: in most of the studies included there is no standardized protocol and no mention of the training for all healthcare providers in visualizing the tip of the catheter using US [43,44]. These two issues are probably very relevant in establishing US performance; in fact, several studies have proved a direct relationship between the number of tips visualized and the training of the operators [27,44]. On the other hand, the tip location using X-ray is not affected by training (there is no date to support this hypothesis). Therefore, it is highly likely that, in the future, US will have a higher chance of successfully locating the tip as soon as well-established protocol and training programs develop.

According to several studies, US is highly efficient for the assessment of tip position during ECC insertion [11], even in low birth weight (LBW) newborns and in SGA [24]. US is also useful after the insertion, for serial assessment [25].

In our meta-analysis, the tip of the catheter was properly visualized with US in a correct position in 87% of the cases, while with X-ray in 77% of the cases (*p* = 0.002), meaning that US performs better when compared to chest X-ray. This higher performance is also clear in the diagnosis of the malposition (US revealed 85.5% of malposition, whereas X-ray showed only 22% of them, *p* = 0.04). US was also found to be faster in tip location when compared to conventional radiology.

The concordance reported between X-ray and US in the tip location of ECCs is 59–100% [1,11,23,24,26,27]. In these studies, 5–25% of ECC tips were in the heart even though they looked to be appropriately positioned on X-ray [27]. This is also very important since it means that conventional radiology can easily and falsely reassure clinicians about the tip location of such devices. The reason behind the high performance of US in the neonatal population probably lies in the fact that neonates have good acoustic windows for examination of the right atrium (RA), superior vena cava (SVC) and inferior vena cava (IVC). Consequently, the relationship between tip and CAJ can be accurately assessed. Additionally, the agreement coefficient between X-ray and US was highest among the lowest birth weight neonates, due to the better penetrance of US in smaller neonates [23]. Thus, the study with US shows a substantial gain of time versus chest X-ray [28]. Another advantage using US is the early detection of complications, such as catheter migration, pleural and pericardial effusion [29].

In some case, however, US cannot correctly detect the tip position of ECCs. The reason for this might include inappropriate training of the operator, the lack of a standardized protocol for tip location, primary malposition into cervical vessels, mammalian vein, hyper-inflated lungs, pneumothorax and distended gaseous abdomen, which may obscure tip localization [23,26]. The use of ECCs is unfortunately associated with a large number of complications, such as CLABSI, thrombosis, phlebitis, occlusion and infiltration [42], cardiac tamponade, myocardial infiltration, pleural effusion, ascites (ECC in inferior vena cava), pericarditis, erosion into pulmonary vessels, paraplegia and myoclonus (tip in ascending lumbar vein) [41]. The reported incidence of ECC-related complications varied from 2.9 to 49.5% [1,26,41], and early or non-elective removal reached 43.27% of cases [11,42]. These complications are, at least in part, related to the malposition of the tip and can thus be prevented especially with an extensive use of US for tip location [2].

Previous studies also found that the insertion site was associated with ECC-related complications, but the results were inconclusive and malposition remains one of the main reasons for the non-elective removal of ECCs [42,45,46,47,48].

In conclusion, our study supports what has been recommended by the recent 2021 INS guidelines. Looking at the data gathered from the available literature, US performs better in tip location when compared to chest X-ray. These data can however underestimate the real power of US, since we do know that the performance of US can be improved with proper training and through the use of a standardized protocol.

In this regard, we would like to quote the recent work by a group of Italian experts, which is a standardized protocol for tip navigation and tip location nicknamed the NeoECHO tip protocol.

This protocol is helpful for real-time tip navigation and tip location, as well as for recurrent tip evaluation, to rule out secondary malposition or complications. The Neo-ECHOTIP protocol includes both real-time US for tip navigation and real-time US for tip location, and it is diversified depending on the insertion site of the vascular access device [8].

*Insertion from the veins of the scalp or from the upper limbs*.

**Tip navigation protocol**:

**Probe: linear hockey stick, 10–14 MHz**.

Acoustic windows: RaPeVA and RaCeVA.

The RaPeVA study helps in the progression of the catheter through the deep veins of the arms [49]. The RaCeVA protocol is useful for the tip navigation of the catheter into the subclavian vein, the brachio-cephalic vein and the SVC, avoiding primary malposition [50].

Tip location protocol:

Probe: small sectorial, 7–8 MHz.

Acoustic windows: at least three windows have been used for the tip location [38,51].

Sub-costal longitudinal view, (bicaval view) for the study of IVC, RA and SVC.Four-chambers apical view, for the study of the four cardiac chambers.Parasternal, long-axis view, for the study of SVC in long axis, RA and azygos vein.

*Insertion from the veins of the lower limbs*.

**Tip navigation protocol**:

**Probe: linear hockey stick, 10–14 MHz**.

Acoustic windows: short- and long-axis view of the femoral vein and the iliac vein.

Tip location protocol:

Probe: small sectorial, 7–8 MHz.

Acoustic windows:Subcostal longitudinal view, for the study of IVC and the RA.

The tip position should be analyzed 24–48 h later to exclude late tip migration and secondary malposition.

Even though this meta-analysis proved the superiority of US it has several limitations:The type of papers included, with only three trials available.Most studies are old and might not reflect the actual ability in this new US era.The studies were performed in an era without a standard protocol for tip location.The training of health care providers was not considered.

## 5. Conclusions

This meta-analysis showed that US is more accurate and reliable in assessing ECC tip when compared to standard X-rays. The data reported might underestimate the benefits in the use of US. This is mainly because most of the studies included in the analysis did not use a standardized protocol for tip location and did not perform proper training of health care providers in the use of US (as is conducted today), both of which are crucial for the performance of US itself.

## Figures and Tables

**Figure 1 diagnostics-13-02850-f001:**
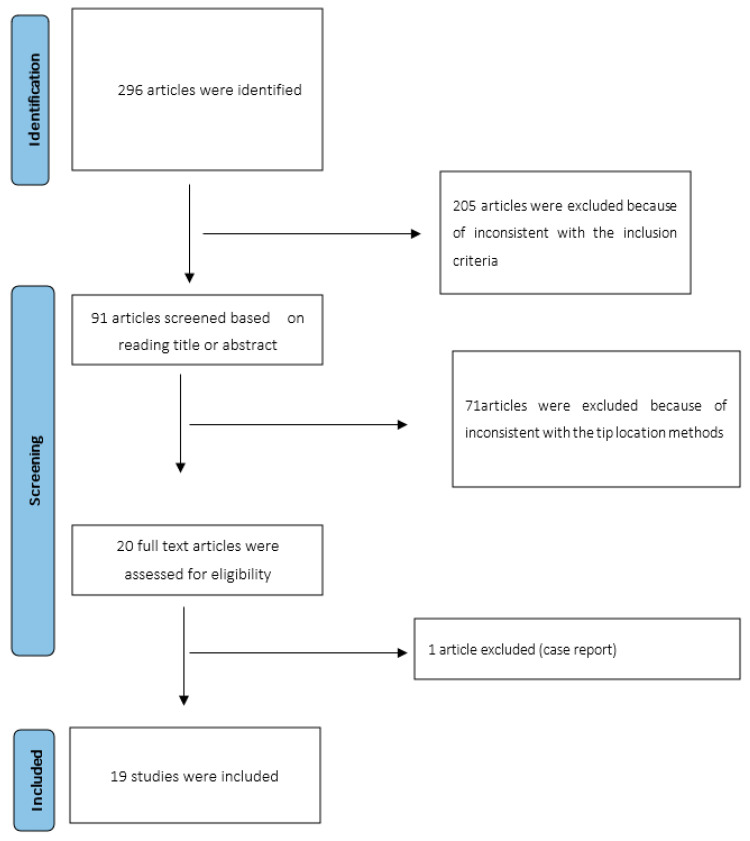
PRISMA flow-chart.

**Figure 2 diagnostics-13-02850-f002:**
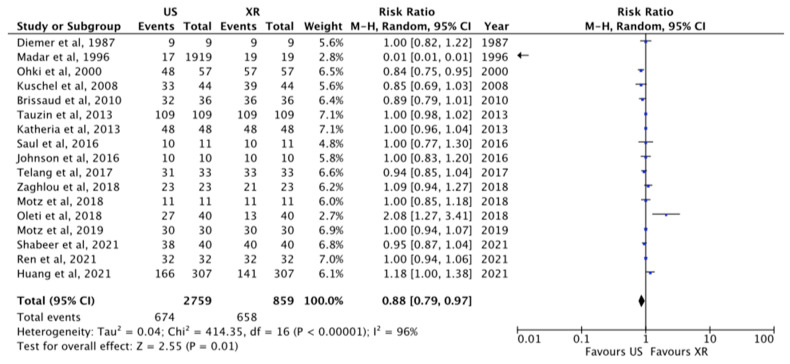
Tip visualization at US and X-ray [1,21,22,23,24,25,26,27,28,29,30,31,32,33,34,35,36]. The central area of the diamond represents the total result of the Meta-analysis; the extremities of the diamond show the CI and the blue square evidenced the amount of the sample reported for each author. The arrow means that the sample size for this paper go beyond the scale used in this figure.

**Figure 3 diagnostics-13-02850-f003:**
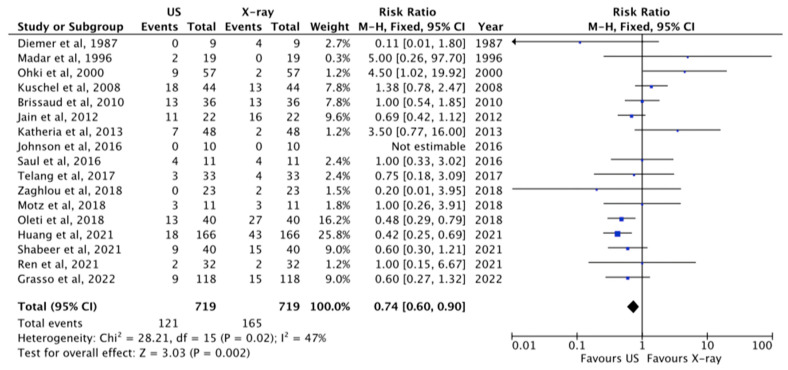
Correct tip position at US and X-ray [1,11,21,22,23,24,25,26,28,29,30,31,32,33,34,35,37]. The central area of the diamond represents the total result of the Meta-analysis; the extremities of the diamond show the CI and the blue square evidenced the amount of the sample reported for each author. The arrow means that the sample size for this paper go beyond the scale used in this figure.

**Figure 4 diagnostics-13-02850-f004:**
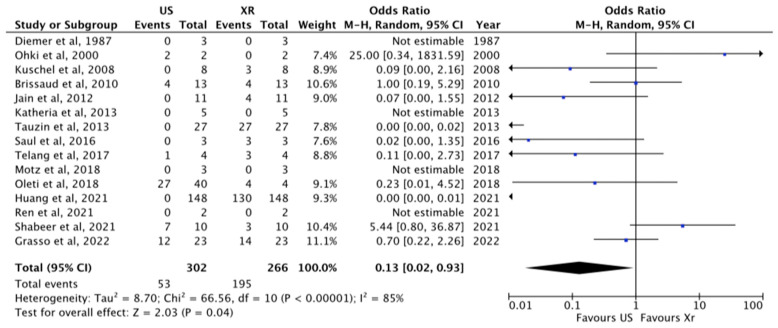
Malposition visualized at US and X-ray [1,11,21,22,24,25,26,27,28,29,31,32,34,35,37]. The central area of the diamond represents the total result of the Meta-analysis; the extremities of the diamond show the CI and the blue square evidenced the amount of the sample reported for each author. The arrow means that the sample size for these papers goes beyond the scale used in this figure.

**Figure 5 diagnostics-13-02850-f005:**
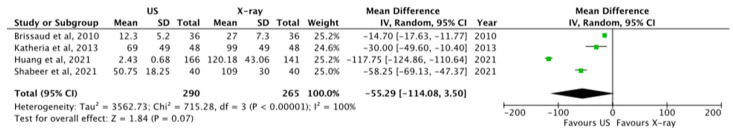
Timing of Insertion and Securing ECCs [21,28,34,35]. The central area of the diamond represents the total result of the Meta-analysis; the extremities of the diamond show the CI and the green square evidenced the amount of the sample reported for each author.

**Figure 6 diagnostics-13-02850-f006:**
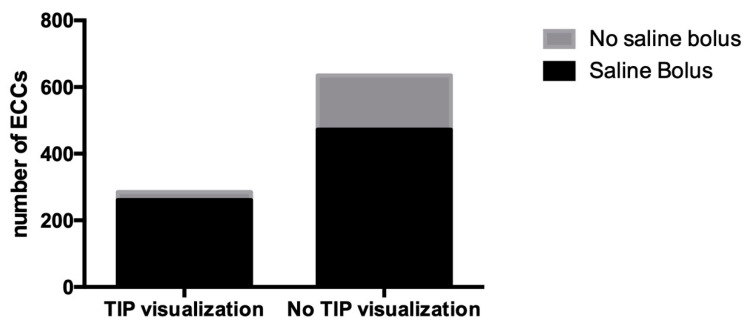
Effect of saline bolus injection.

**Table 2 diagnostics-13-02850-t002:** GRADE evidence profile for the present meta-analysis.

Quality Assessment							No. of Patients		Quality	
No. of Studies	Study Design	Risk of Bias	Inconsistency	Indirectness	Imprecision	Other Considerations	Cases	Controls	Relative (95% CI)	
Tip visualization at US and X-ray							US	X-ray		
17	OS	Moderate	Considerable	Not serious	Serious	None	2759	859	RR 0.88 [0.79–0.97]	⊗⊗OO LOW
Correct TIP position at US and X-ray							US	X-ray		
17	OS	Moderate	Moderate	Not serious	Serious	None	719	719	RR 0.74 [0.60–0.90]	⊗⊗⊗O MODERATE
Malposition visualized at US and X-ray							US	X-ray		
15	OS	Moderate	Substantial	Not serious	Serious	None	302	266	RR 0.13 [0.02–0.93]	⊗⊗OO LOW
Timing of TIP location for US and X-ray							US	X-ray		
4	OS	Moderate	Considerable	Not serious	Serious	None	290	265	MD −55.29 [−114.08, 3.50]	⊗⊗OO LOW

These symbols (⊗) are the iconographic final expression of the level of Quality, depending from kind of study (observational, RCT), bias, inconsistency, indirectness, imprecision (I2). One cross means very low quality, two crosses low quality, three moderate and four crosses high quality.

## Data Availability

The data will be made available, in addition to study protocols, the statistical analysis plan. Proposals should be submitted to vito.dandrea@policlinicogemelli.it.

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
